energy/power and energy systems

air-conditioning heat pump, electric vehicles, combustion characteristics, blow-off phenomenon, radiation flux

**Authors for correspondence:**
Jun Fang
e-mail: fangjun@ustc.edu.cn
Ran Tu
e-mail: turan@hqu.edu.cn

[†]These authors, as the co-first authors, contributed equally to this work.

# Experimental investigation on combustion characteristics of flammable refrigerant R290/R1234yf leakage from heat pump system for electric vehicles

Kang Li[1,2,†], Jingwu Wang[3,†], Shuxian Luo[1], Zhenzhen Wang[1], Xuejin Zhou[4], Jun Fang[5], Lin Su[1] and Ran Tu[4]

[1]School of Energy and Power Engineering, and [2]Key Laboratory of Multiphase Flow and Heat Transfer in Shanghai Power Engineering, University of Shanghai for Science and Technology, Shanghai 200093, People's Republic of China
[3]Hefei Institute for Public Safety Research, Tsinghua University, Hefei, Anhui 230601, People's Republic of China
[4]College of Mechanical Engineering and Automation, Huaqiao University, Xiamen, Fujian 361021, People's Republic of China
[5]State Key Laboratory of Fire Science, University of Science and Technology of China, Hefei, Anhui 230026, People's Republic of China

RT, 0000-0002-2697-196X

Concerning the issues regarding driving mileage reduction for electric vehicles (EVs) in cold climates, a heat pump system with low global warming potential refrigerant R290/R1234yf is employed as one of the promising solutions. Different from the widely used mobile refrigerant R134a, R290 and R1234yf are both flammable or explosive. The application of R290/R1234yf in the mobile heat pump system is hindered by unexpected refrigerant leakage with the existence of fire and explosion risk. In this study, the combustion characteristics of R290/R1234yf in a potential leakage process from an air-conditioning heat pump system for EVs were investigated. Firstly, thermodynamic behaviours of R290/R1234yf used in a typical heat pump system were analysed based on a special experimental facility designed for EVs. Then the leakage and combustion characteristics of R290/R1234yf including flame shape, temperature, radiation etc. were obtained by the experimental method under different initial temperature and mass flow rate conditions. It was found that R290/R1234yf leaked is difficult to ignite at low temperatures,

while the blow-off phenomenon of the jet flame would occur at high temperature with high leakage mass flow rate. In addition, the results showed that combustion intensity would be enhanced by the leakage mass flow rate between 30 and 60°C. These results could provide guidance for fire detection and rescue system design for new energy vehicles.

## 1. Introduction

Electric vehicles (EVs) have drawn increasing worldwide attention as an alternative to internal combustion engine vehicles (ICEVs), owing to their lower energy consumption and almost zero carbon emission. Different from traditional ICEVs, EVs could not use waste heat from the engine for cabin warming in the cold climate. Meanwhile, additional energy consumption to maintain cabin comfort would have significant influence on driving mileage. The positive temperature coefficient (PTC) element, widely used in the car industry, is considered as a convenient solution to provide heat. However, using PTC would cost much more energy from the battery, resulting in approximately 50% reduction in driving mileage in winter [1,2]. Therefore, heat utilization from air source by an air-conditioning heat pump (ACHP) system is proposed and proved to increase by 20% driving mileage compared with PTC [3,4]. Currently, R134a heat pump systems are widely studied and tested in EVs which showed a quite good performance at an ambient temperature above −10°C [5,6]. But when below −10°C, R134a heat pump could not provide sufficient heat for its special refrigerant thermo-physical properties [7,8]. Furthermore, high global warming potential (GWP) refrigerant will be phased down gradually after January 2019 due to the Kigali Amendment, which more than 170 counties agreed and signed in 2016 [9,10]. It indicated that R134a would be replaced by low GWP refrigerants in the near decades in EVs with great probability (table 1).

To meet both challenges in cabin heating for cold climate and refrigerant substitution, many kinds of heat pump systems with low GWP refrigerant are proposed. The $CO_2$ heat pump system is widely considered as a promising solution, which has shown an excellent heating performance in an extremely cold climate. A previous study showed that a $CO_2$ heat pump could achieve a coefficient of performance (COP) of 3.1 and heating capacity of 3.6 kW at −20°C for both cabin and ambient temperature in EVs [11,12]. Whereas, applying $CO_2$ as a refrigerant in the air-conditioning (AC) system in EVs could not achieve a satisfactory cooling performance, especially at extremely high temperatures (above 30°C) [13,14]. Besides, the $CO_2$ molecule is much smaller than other hydrocarbons (HC)/hydrofluorocarbons (HFC) refrigerant molecules, consequently, highly pressurized $CO_2$ might leak from the aluminium tubes of the ACHP system [15]. This leads to serious concern regarding the reliability and efficiency of $CO_2$ heat pump systems at complicated operating conditions in EVs, which still needs further investigation.

R290 and R1234yf are the other two options as the alternative refrigerants in EVs. R1234yf was initially designed to replace R134a and has already been applied in some ICEVs [16], whereas R1234yf is flammable which may cause fire or explosion in the case of accidents. On the other hand, the R290 heat pump system in EVs was proposed and it might be a solution for its good thermodynamic properties and low GWP [17,18]. The R290 heat pump system has similar components and structure to the typical R134a heat pump system, as operation pressure of R290 is at the same level as R134a. Research into the R290 heat pump system in EVs is still at the early stage, but as a substitution of R22 in household air-conditioning system, R290 has been widely investigated for decades [10,19–21]. In the heat pump system, R290 has shown a similar heating performance with $CO_2$ above −20°C and larger volume cooling capacity compared with R134a [17,22]. Although R290 has many advantages in replacing R134, the flammable and explosive characteristics are the main hindrance to its promotion as a heat pump system for EVs.

For further application of R290 and R1234yf refrigerant in EVs, leakage and combustion risk assessment need to be clarified. R290 and R1234yf were classed as A3 and A2 L flammable refrigerant, respectively, by the American society of heating, refrigerating and air-conditioning engineers (ASHARE) [23]. Under normal temperature conditions, R290 could be easily ignited causing potential fire or explosion hazard in R290 from the ACHP system when the unpredictable leaked R290 reaches a lower flammable limit (LFL). Previous study on leakage behaviours of R290 from household air conditioning showed that the fire hazard could probably occur in the early stage of the leakage process near the leak source [24]. Feng *et al.* [25,26] have studied the combustion and explosion characteristics of R290 and R1234yf near LFL, which focused on the effect of gas disturbance and combustion intensity, but without considering the effect of other initial conditions such as gas temperature. Clodic & Jabbour have reported the burning rates of R290 using a tube experimental

**Table 1.** Nomenclature.

| | |
|---|---|
| EVs | electric vehicles |
| ICEVs | internal combustion engine vehicles |
| PTC | positive temperature coefficient |
| GWP | global warming potential |
| COP | coefficient of performance |
| AC | air conditioning |
| HC | hydrocarbons |
| HFC | hydrofluorocarbons |
| ACHP | air-conditioning heat pump |
| LFL | lower flammable limit |
| TXV | thermal expansion valves |
| IHEX | indoor heat exchanger |
| OHEX | outdoor heat exchanger |

method for household air conditioner [27]. We noted that there are relative studies on the combustion characteristics of R290 leakage for household ACs, but little research into EV. Further, a series of combustion experiments were carried out by Zhang *et al.* [28] on laboratory scale, which indicates that the over-pressure leaked R290 is not sufficient to cause AC system damage, but if R290 was ignited during the leak, the system would quickly be burned out.

Compared with the risk assessment and combustion investigation of flammable refrigerants for the household AC system, literature about R290 or R1234yf combustion behaviour during the leakage process from the ACHP system in EVs or ICEVs is still limited. In this paper, a newly designed ACHP system, applied in a type of EV, was introduced. Based on this system, thermodynamic characteristics were studied using R290 and R1234yf as the alternative refrigerant of R134a. Then, a comprehensive experimental study using R134a/R290/R1234yf with various initial temperatures and volume flow rates were conducted under operation conditions. Typical combustion and thermal dynamics parameters including flame height and centreline temperature of flame under various conditions were analysed. The ignition temperature and radiation flux were also recorded to investigate the combustion behaviour of the R290/R1234yf flame. Finally, the combustion behaviour of these flammable refrigerants was compared and followed with some conclusive remarks. The data obtained could provide a reference for the safe design of EVs with an R290/R1234yf ACHP system.

# 2. Experiments

## 2.1. Thermodynamic analysis

According to the ASHRAE Standard mentioned above, hydrocarbon refrigerants R290 and R1234yf are flammable and explosive, but also environmentally-friendly refrigerants with GWP values of 11 and 4, respectively. With reference to the ASHRARE Standards Committee [29], other thermodynamic properties are listed in table 2 thoroughly.

## 2.2. Experimental heat pump system set-up

Due to the greenhouse effect, the refrigerant R134a has forcibly been replaced in the automobile industry and the refrigerant R290 will be the alternative environmentally-friendly medium. To figure out the risk of the R290 use for an automobile ACHP system, a three heat exchanger–ACHP system with functions of cooling and heating was designed, as shown in figure 1. This is a calculated model of an R290 vapour compression system, which includes components of two electronic valves, two thermal expansion valves (TXV), two indoor heat exchangers (IHEX), two mass flow meters, outdoor heat exchanger (OHEX), compressor, gas–liquid separator, PTC, etc. This system can realize cooling or heating

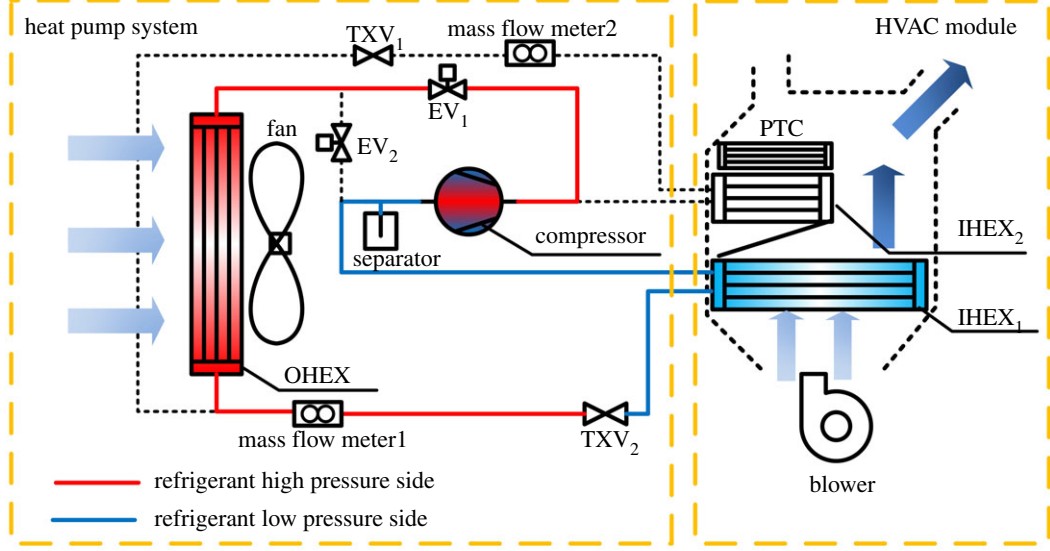

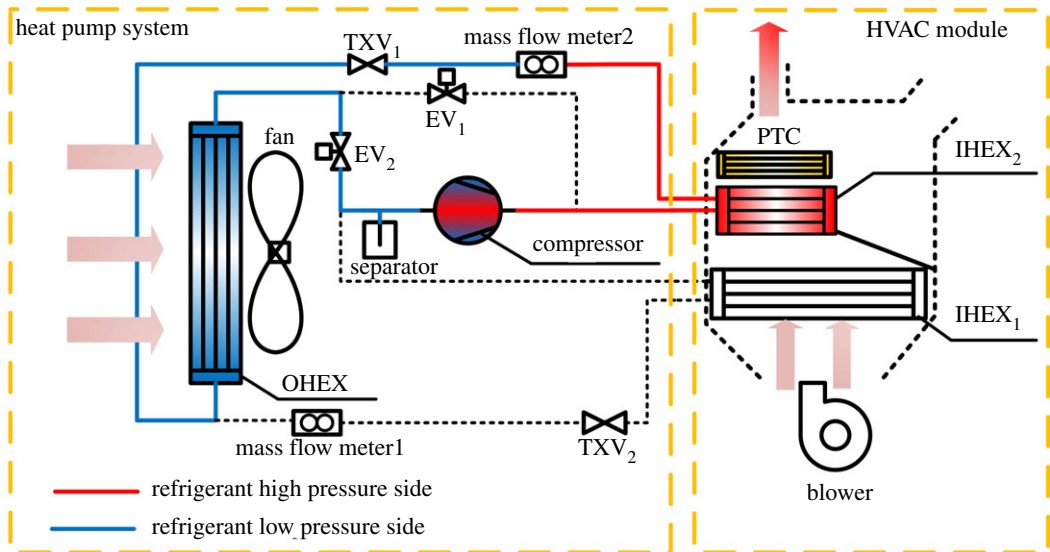

OHEX: outside heat exchanger       IHEX: inside heat exchanger
TEV: thermal expansion             EV: electrical valve

**Figure 1.** A schematic of newly designed ACHP system for EVs.

**Table 2.** Characteristic parameters of refrigerants use in the ACHP system.

| item | R290 | R1234yf | R134a |
|---|---|---|---|
| molecular mass (kg kmol$^{-1}$) | 44.096 | 114 | 102 |
| critical temperature (°C) | 134.6 | 96 | 101.1 |
| critical pressure (MPa) | 4.23 | 3.382 | 4.059 |
| GWP | 11 | 4 | 1430 |
| safety group | A3 | A2 L | A1 |
| formula | $C_3H_8$ | $C_3H_2F_4$ | $C_2H_2F_4$ |

functions by switching two solenoid valves to change the refrigerant flow chart. In the cooling mode, electronic valve 1 is used to connect the pipeline between the compressor and the outdoor heat exchanger, and the indoor heat exchanger 1 is used to absorb redundant heat from the passenger

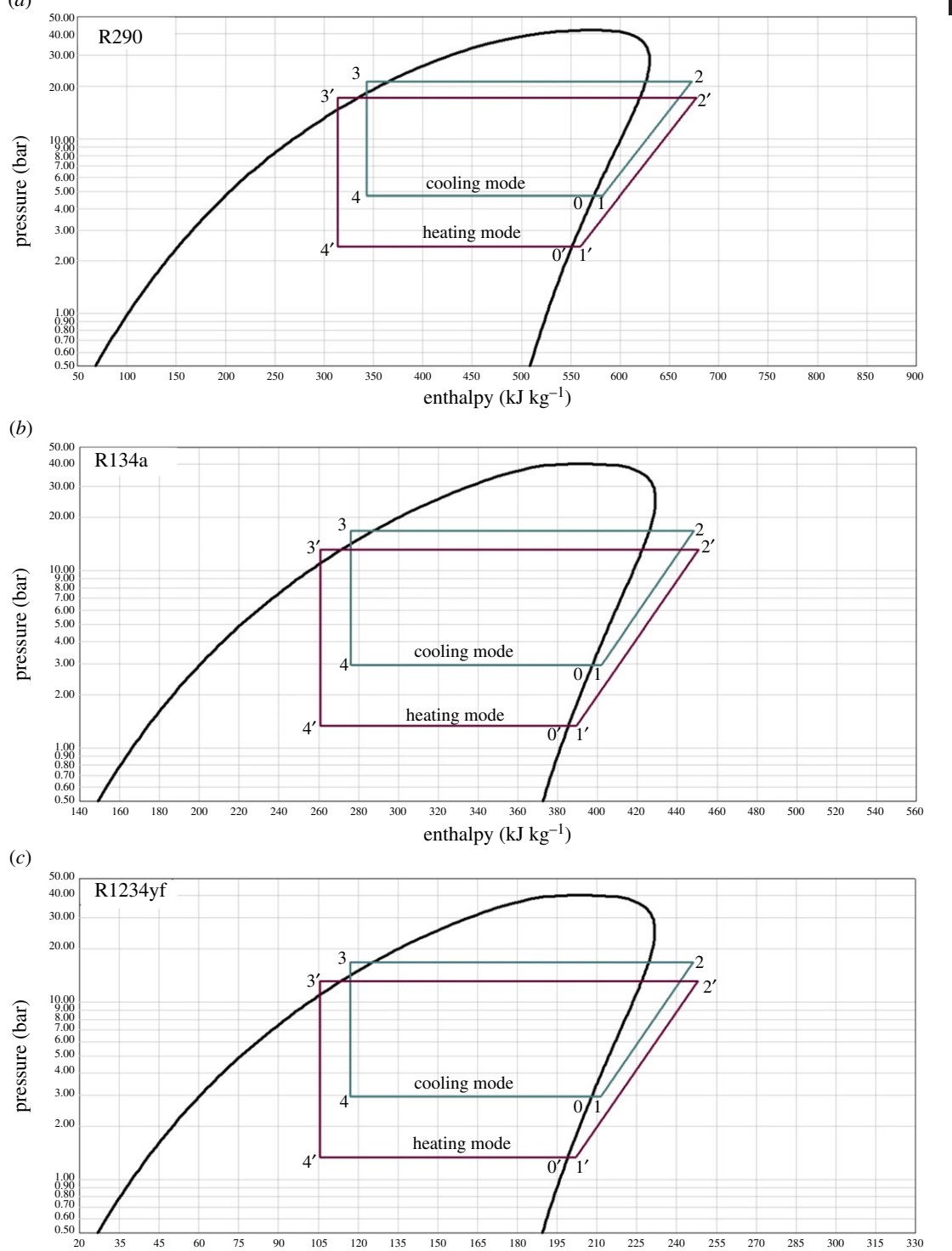

**Figure 2.** Log P–h diagrams for three refrigerants: (a) R290, (b) R134a and (c) R1234yf, respectively.

compartment. On the contrary, for the heating mode, electronic valve 2 and indoor heat exchanger 1 start to work, and the PTC heat exchanger works as a supplementary heat source during the extreme condition.

A comparison chart of the related three different refrigerants under the same conditions of evaporation and condensation is shown in figure 2 by NIST Refprop 8.0 [30] for a better understanding. Each chart is composed of four processes as the pressure–enthalpy diagram for both cooling and heating modes: $1 \rightarrow 2$, compression in the compressor; $2 \rightarrow 3$, condensation in the condenser; $3 \rightarrow 4$, throttling process through the throttle valve; $4 \rightarrow 1$, evaporation in the evaporator. Detailed specifications of the experimental conditions are shown in tables 3 and 4.

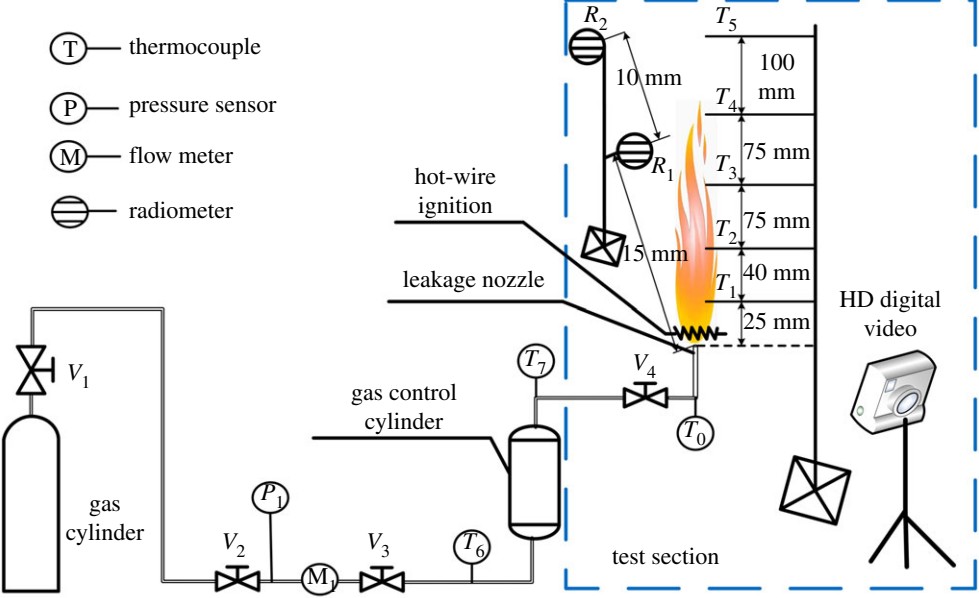

**Figure 3.** Experiment facility for combustion behaviour of jet flame in leakage.

**Table 3.** Cooling test conditions.

| parameters | R290 | R124yf | R134a |
|---|---|---|---|
| environment temperature (°C) | 38 | 38 | 38 |
| evaporating temperature (°C) | 0 | 0 | 0 |
| evaporating pressure (MPa) | 0.472 | 0.314 | 0.292 |
| condensing temperature (°C) | 60 | 60 | 60 |
| superheating temperature (°C) | 5 | 5 | 5 |
| subcooling temperature (°C) | 7 | 7 | 7 |
| condensing pressure (MPa) | 2.124 | 1.636 | 1.688 |
| compressor discharge temperature (°C) | 87.06 | 64 | 72.00 |

**Table 4.** Heating test conditions.

| parameters | R290 | R124yf | R134a |
|---|---|---|---|
| environment temperature (°C) | −10 | −10 | −10 |
| evaporating temperature (°C) | −20 | −20 | −20 |
| evaporating pressure (MPa) | 0.242 | 0.150 | 0.132 |
| condensing temperature (°C) | 50 | 50 | 50 |
| superheating temperature (°C) | 5 | 5 | 5 |
| subcooling temperature (°C) | 7 | 7 | 7 |
| condensing pressure (MPa) | 1.706 | 1.298 | 1.313 |
| compressor discharge temperature (°C) | 83.26 | 62.00 | 71.67 |

## 2.3. Experimental combustion system set-up

The ignition energy and flame behaviour of R290/R1234yf were investigated by a jet flow combustion system, as shown in figure 3. Components of the system include gas control cylinder, flow meter, radiometers, hot-wire ignition, leakage nozzle, etc. The temperature and pressure data of four survey points are measured by thermocouples and pressure sensors within an accuracy ±0.25% and a response time of less than 0.2 s.

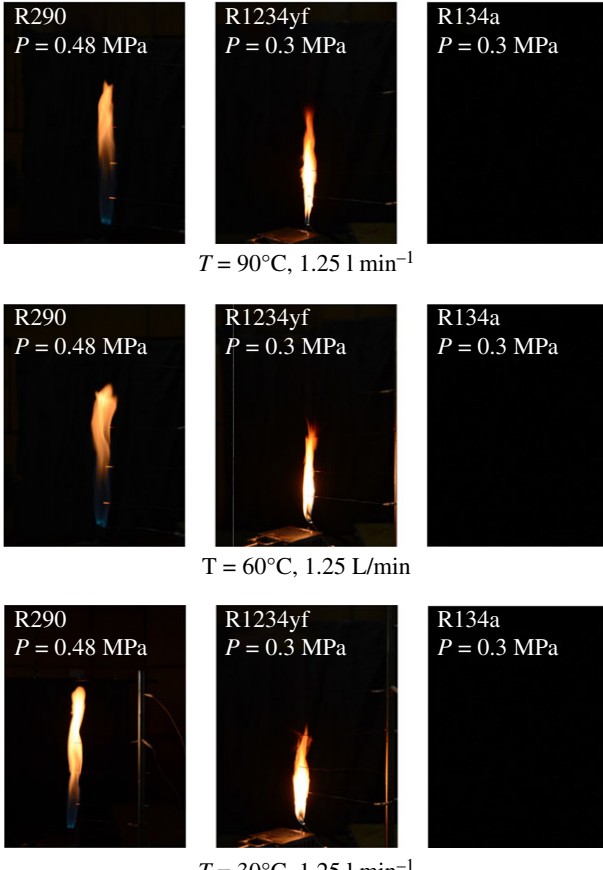

**Figure 4.** Flame shape comparison of R290, R1234yf and R134a.

**Table 5.** Combustion condition.

| item | R290 | R1234yf | R134a |
|---|---|---|---|
| pressure (MPa) | 0.48 | 0.3 | 0.3 |
| volume flow rate (l min$^{-1}$) | 0.5/0.75/1/1.25/1.5 | 0.5/0.75/1/1.25/1.5/1.75/2/2.25 | 0.5/0.75/1/1.25/1.5 |
| gas temperature (°C) | 10/30/60/90 | 30/60/90 | 30/60/90 |

To supervise and record the thermal radiation of the flame in different directions, two radiometers were employed in this experiment. The K-type thermocouple array was used to measure instantaneous temperatures in different flame heights (from 25 to 100 mm) for the three flammable refrigerants. To simulate that the refrigerant leaked from automobile ACHP system crashed lines or corroded heat exchangers, the leakage nozzle of 1 mm diameter is arranged for the scenery imitation. The HD digital video camera was used to obtain the image data. Refrigerant pressure at the transport line was recorded by $P1$, while temperature was recorded by $T6$. Temperature variation of refrigerant parameters in the nozzle front and back were recorded by $T0$ and $T7$. Combustion conditions of R290 and R1234yf are shown in table 5.

Moreover, the R1234yf flames were ignited by a small flame with a volume flow rate of propane (R290) of 0.02 l min$^{-1}$, to ensure the burning process remained steady and consistent.

# 3. Results and discussion

## 3.1. Refrigerant combustion comparison

Comparisons of flame shapes with three types of refrigerants under the same volume flow rate 1.25 l min$^{-1}$ are summarized in figure 4. The initial gas temperatures are 30°C, 60°C and 90°C, respectively (here, we cool

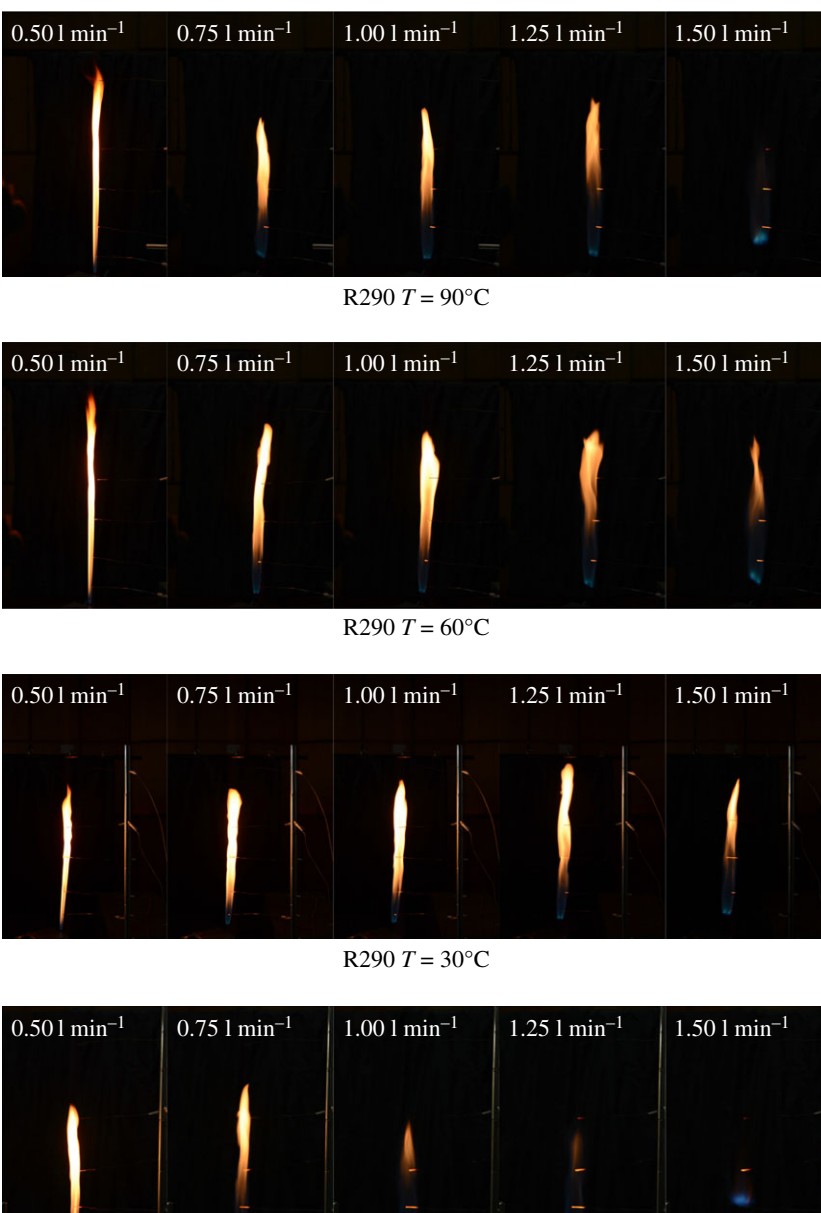

**Figure 5.** Flame variations of R290 under different conditions.

the refrigerant gas to 10°C with an ice cube at first and then heat the gas cylinder with electric heating tape to the preset initial temperature). The pressure of refrigerant R290 is 0.48 MPa, while those of R1234yf and R134a are 0.3 MPa. R134a is hard to ignite under normal operating conditions, as shown in figure 4. The flame of R290 or R1234yf varies with the change of gas temperature, and it is evident that the flame of R290 is larger than that of R1234yf. Under volume flow rate 1.25 l min$^{-1}$, R1234yf had the highest flame height when gas was heated to initial an temperature of 90°C by observation. It is the competitive result by effects of burning enhancement by temperature, flow turbulence or destabilization by exit velocity. Among three temperature conditions, the flame height of R290 is approximately unchanged, while combustion intensity seems to be enhanced with the increasing gas temperature from flame colour. As the temperature rises, the colour of the flame ranges from red and orange to yellow and white.

## 3.2. Effect of the gas temperature and volume flow rate

Flame morphology variations under different flow rate and temperature conditions of R290 are shown in figure 5. When the R290 with a volume flow rate of 0.5 l min$^{-1}$ was ignited, a relatively slender flame

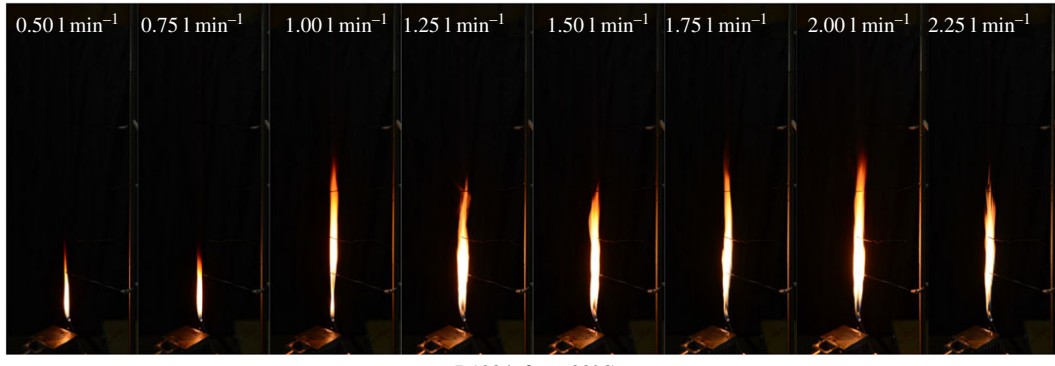

| 0.50 l min⁻¹ | 0.75 l min⁻¹ | 1.00 l min⁻¹ | 1.25 l min⁻¹ | 1.50 l min⁻¹ | 1.75 l min⁻¹ | 2.00 l min⁻¹ | 2.25 l min⁻¹ |

R1234yf $T$ = 30°C

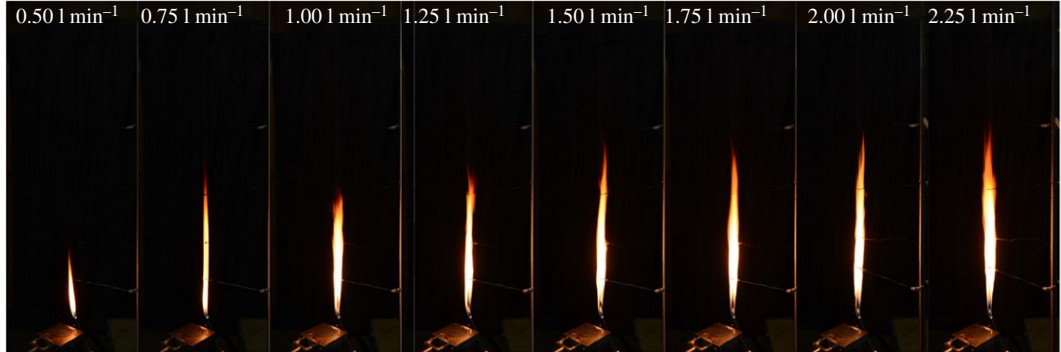

R1234yf $T$ = 60°C

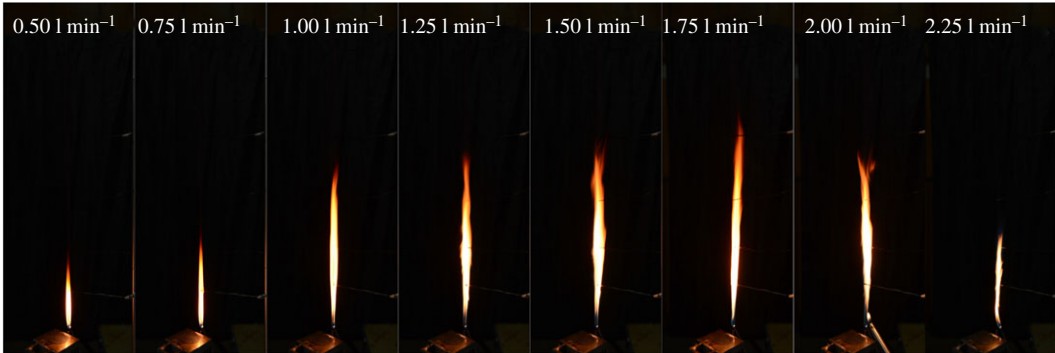

R1234yf $T$ = 90°C

**Figure 6.** Flame images of R1234yf under various initial temperatures.

appeared at the leakage nozzle, and the higher refrigerant temperature led to a slenderer flame, which is due to the enhanced combustion intensity shrinking the burning zone. With the increase in the volume flow rate for the same temperature, the flame's yellow region (soot particle emission) showed an almost opposite trend to flame height, and this phenomenon would be more obvious as the temperature increased. This is because with the increasing flow rate, the mixing of air and gas would be more sufficient, especially near the exit (at the flame bottom), which reduced the yellow part of the flame. By experimental video information, it was found that the R290 burned vigorously under initial temperature of 10°C, and the flame showed more blue colour, indicating full combustion. Besides, under the condition of 90°C and 1.5 l min⁻¹, R290 almost cannot maintain combustion regime accompanied by a hissing sound.

Considering the difficulty of igniting the refrigerant R1234yf, the R290 flame was used to ignite the R1234yf gas flow in this experiment. Due to the R1234yf molecular structure with a higher carbon ratio than R290, the combustion of R1234yf produced black smoke, accompanied by a pungent smell. The flame of R1234yf was notably thinner than R290 at 90°C, and it was apparent that the combustion process looked smoother as shown in figure 6. It can be observed that the flame length is significantly increasing when the concentration is increased to approximately 2.0 l min⁻¹. However, taking the

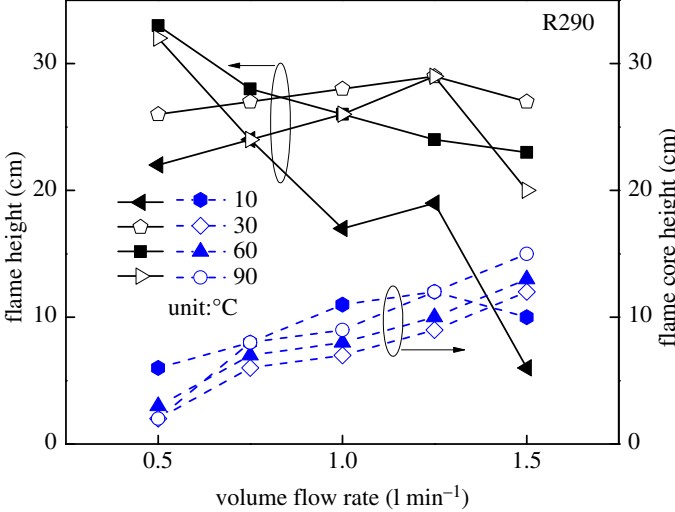

**Figure 7.** Flame height of R290 (solid line: flame height; dash line: flame core height).

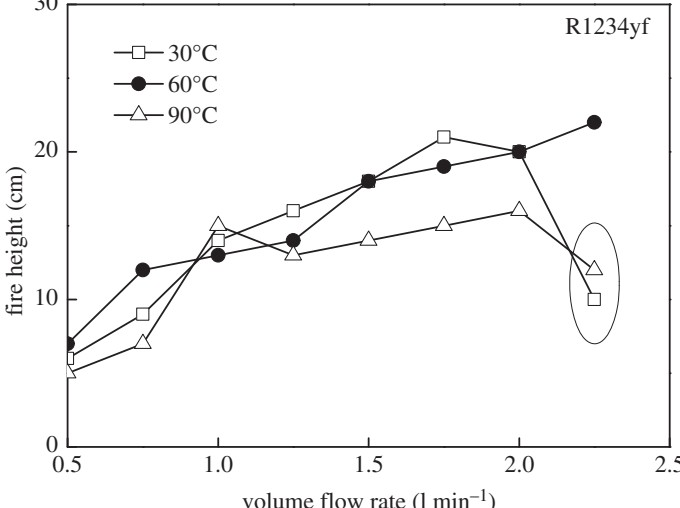

**Figure 8.** Flame height of R1234yf under various conditions.

R1234yf with 2.5 l min⁻¹ as an example, an unsteady flame is formed at a temperature of 90°C, which is opposite to the condition of 30°C /60°C (remains steady). Considering the volume flow rate effects on the R290 and R1234yf, it is easy to note that the R1234yf flame height is shorter than R290 in smaller flux.

## 3.3. Summary of flame height

The detailed flame height and flame core height of R290 under various refrigerant gas temperatures and flow rate are summarized as shown in figure 7.

With the increasing flow rate, flame core height and flame height present different rules. As the volume flow rate increases from 0.5 to 1.25 l min⁻¹ at a temperature of 10°C, the flame height decreases and the flame core height increases. However, both of the two heights begin to fall sharply in the case of the volume flow rate exceeding 1.25 l min⁻¹, when it is observed that R290 gas burns fiercely under large flux and low-temperature condition, and the strong turbulence in burning would consequently reduce flame heights. Therefore, to resolve the concern of refrigerant explosion caused by leakage in the automobile ACHP system, the leakage rate should be considered. Moreover, as indicated in figure 7, the effects of gas temperature on flame core height are insensitive without a monotonic relationship.

Further for R1234yf gas fire as shown in figure 8, flame height increases consistently from 0.5 to 2.0 min⁻¹, especially in the case of 60°C. Once the volume flow rate exceeds 2.0 min⁻¹, the height of

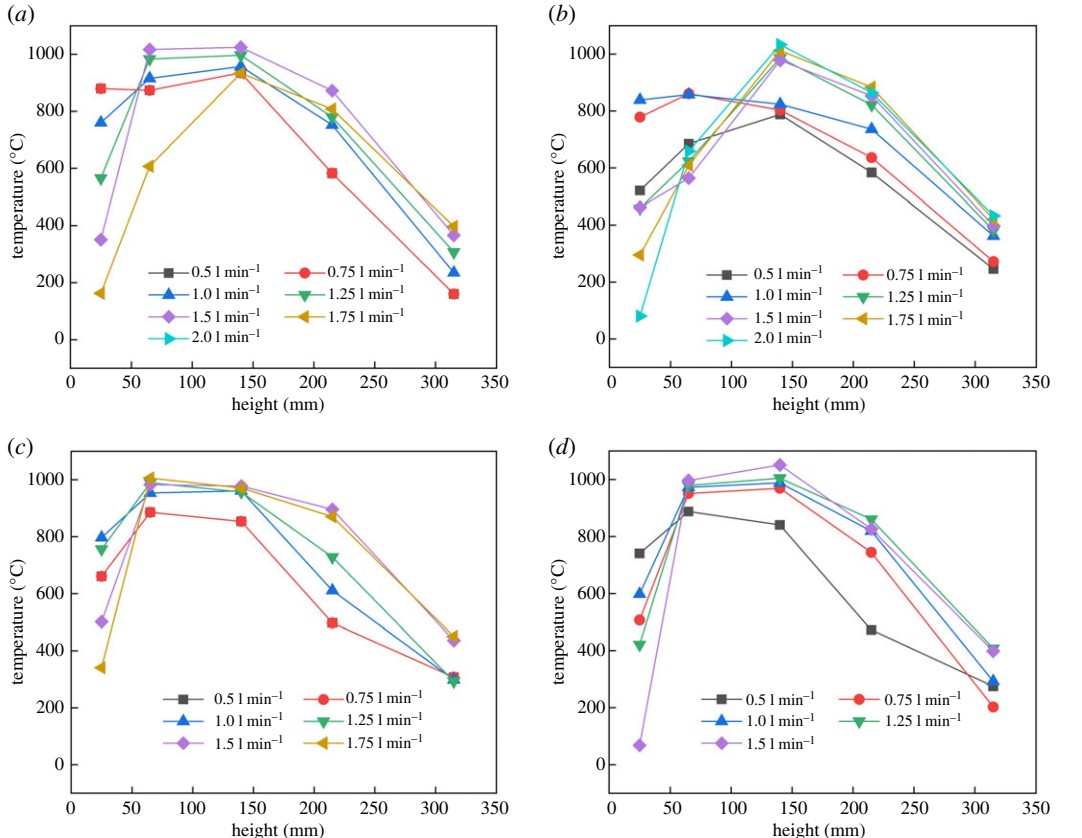

**Figure 9.** Flame temperature of R290 alone centreline in burning under simulated leakage. Leakage temperature: (*a*) 10℃, (*b*) 30℃, (*c*) 60℃ and (*d*) 90℃.

the flame begins to decrease for low and high temperature conditions (30 and 90℃) attributed to the unsteady burning.

## 3.4. Summary of flame temperature

Flame temperatures of R290 and R1234yf in different positions are illustrated in figures 9 and 10, respectively. Figure 9 shows that the flame of R290 can reach the highest temperature while height is between 100 and 150 mm, due to the combustion core zone where the burning reaction is concentrated.

Two different combustion phenomena between R290 and R1234yf mentioned in figures 5 and 6 also lead to differences in flame temperature. As shown in figure 10, for lower R1234yf gas temperature, the flame temperature increases at first, then drops after the second thermocouple temperature point. But for higher R1234yf gas temperature, the fire temperature keeps reducing along with fire height due to the variation of the main combustion reaction zone. It is suggested that the main burning reaction zone would move forward (towards the exit) at higher temperature due to the faster combustion velocity, leading to a lower turning point, as shown in figure 10, and this effect is highly evident for R1234yf gas. It should be noted that, although R290 and R1234yf have an unequal flame, both of them almost have the same maximum flame temperature.

## 3.5. Summary of R290 ignition temperature

The relationship between ignition temperature and volume flow rate was investigated during the experimental study. With the change of the volume flow rate and gas temperature, the ignition temperatures of R290 are shown in figure 11.

As the refrigerant gas temperature increased, it could be seen that the ignition temperatures were usually above 600℃ and remained approximately steady for 10, 30 and 60℃ conditions. But for higher temperature such as 90℃, the ignition became easier except for the bigger volume flow rate.

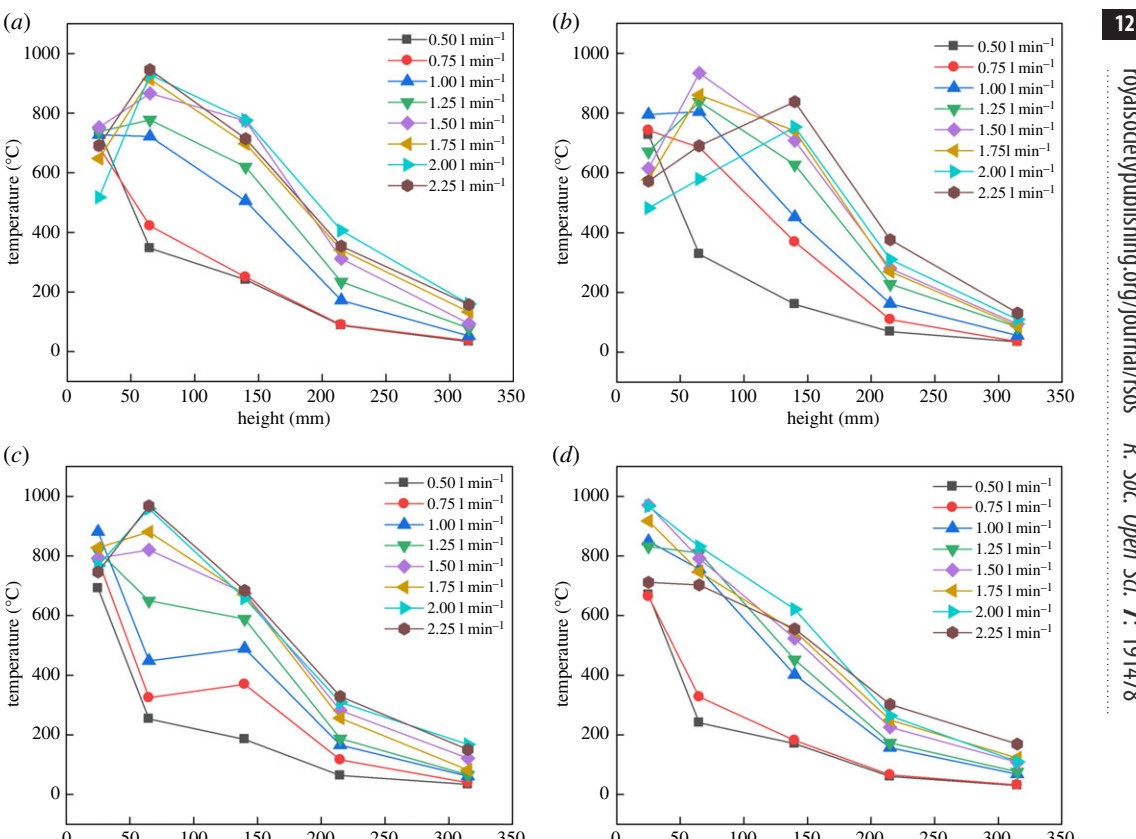

**Figure 10.** Flame temperature of R1234yf in burning under simulated leakage. Leakage temperature: (*a*) 20℃, (*b*) 30℃, (*c*) 60℃ and (*d*) 90℃.

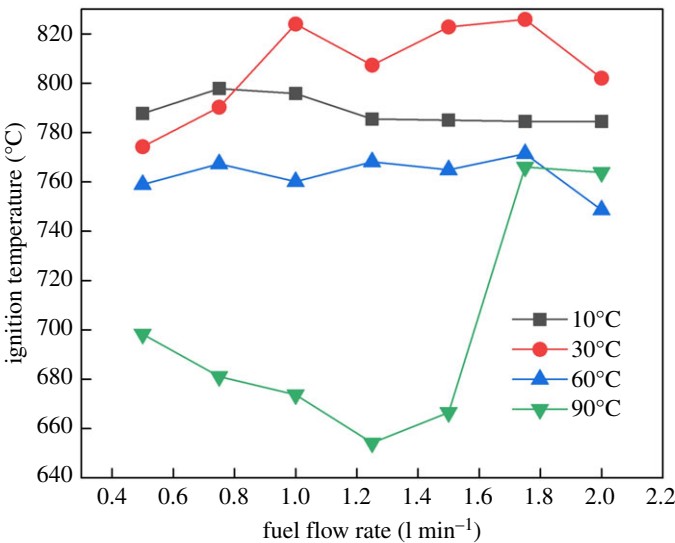

**Figure 11.** Influence of initial temperature and flow rate on ignition temperature of R290.

This phenomenon is attributed to the extra enthalpy by initial temperature which assisted the combustion and ignition process. But on the other hand, the larger flow rate would strengthen the destabilization of the gas mixture to disturb ignition. Therefore, the straightforward conclusion proposed here is that controlling ignition energy in the heat pump system could greatly reduce the danger of burning by means of the actual test phenomena.

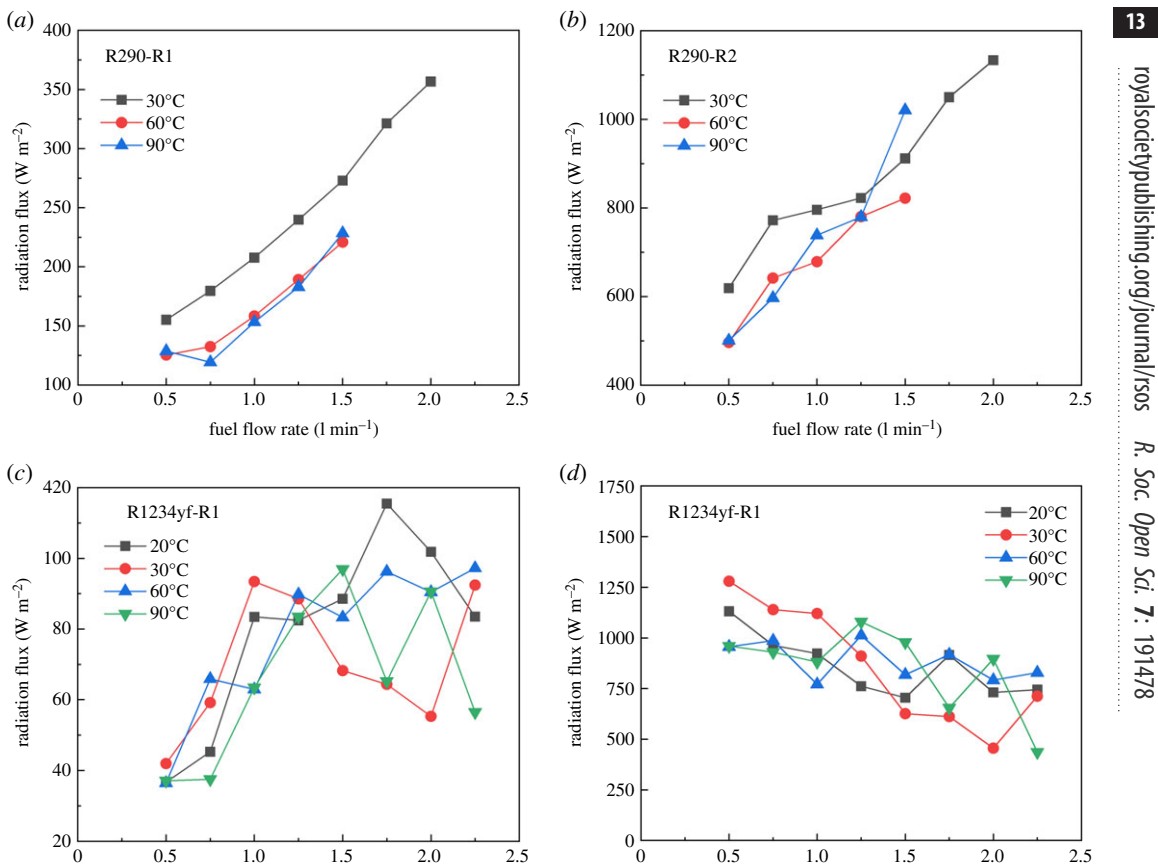

**Figure 12.** Radiation at two positions in various conditions of R290 and R1234yf flame. (*a*) R290-R1, (*b*) R290-R2, (*c*) R1234yf-R1 and (*d*) R1234yf-R2.

## 3.6. Summary of radiation flux

Radiation heat flux was also measured at different leakage volume flow rates and temperatures of R290/ R1234yf as shown in figure 12. Figure 12*a* expresses the radiation flux collected by thermal radiometer 1 (figure 3). It can be seen that the radiation flux at 30°C is higher than that at other temperatures. This is because at lower temperatures, the flame is near the leakage nozzle as shown in figure 5. With the increasing flow rate, the radiation fluxes at various temperatures have an upward trend. Identical trends can be obtained in figure 12*b* for radiometer 2, while the thermal radiation value increases to a larger extent. Compared with the R290, the radiation flux of R1234yf has relatively more complex fluctuations. Figure 12*c* indicates that the heat flux value by thermal radiometer 1 used in the R1234yf flame increases before $1.25\,\mathrm{l\,min^{-1}}$. However, the data collected by the thermal radiometer 2 shows a unique downward trend in all the experimental conditions.

## 4. Concluding remarks

In this paper, combustion characteristics of R290/R1234yf in a simulated leakage process from an ACHP system were studied experimentally. Based on the heat pump system and combustion facility with R290/ R1234yf proposed, the flame shape, flame temperature, ignition energy and radiation flux under the conditions of different volume flow rate and gas temperature were interpreted and analysed. The results are summarized as follows:

(1) Compared with the mass flow rate, the gas temperature showed little effect on flame height of R290, while the combustion intensity was enhanced with high initial temperature. The flame from R290 burning was wider than R1234yf for the same initial condition in general. The flame physical morphology obtained here could provide guidance for fire hazard evaluation and rescue of vehicles.

(2) It was found that the R290 under initial temperature of 10°C burned vigorously, and for 90°C, $1.5 \, \text{l min}^{-1}$ condition, R290 was almost blown out. On the other hand, R1234yf could not be ignited without flaming fire, and was also accompanied by black smoke and a pungent smell. Moreover, flame height showed a positive relation with the refrigerant volume flow rate in general.

(3) Ignition temperatures were usually above 600°C for R290 and kept approximately steady except for the high initial gas temperature condition. Further, radiation flux increased significantly with enlarged flow rate for R290, which is different from the results of R1234yf. These tendencies obtained were believed to be useful for early fire detection and prevention system design inside new energy cars.

Further work will be conducted on leakage combustion behaviours of other refrigerants, e.g. R1234zd and R1234ze.

Data accessibility. We uploaded the experimental data as electronic supplementary material.

Authors' contributions. K.L., J.W., R.T. and J.F. contributed to the initial idea and developed the experimental platform. S.L., Z.W. and L.S. designed the study and performed the experiments. X.Z. and K.L. collected and analysed the data. K.L. and J.W. wrote the manuscript, which was reviewed again by all authors. All authors gave final approval for publication.

Competing interests. The authors declare no competing interests.

Funding. This study was funded by National Nature Science Foundation of China (grant nos 51876130 and 51906158), The Open Project Program of State Key Laboratory of Fire Science (grant nos HZ2018-KF03 and HZ2018-KF09), Anhui Provincial Natural Science Foundation (grant no. 1908085QE205), Shanghai Sailing Program (grant no. 18YF1417900), Capacity Building Plan for some Non-military Universities and Colleges of Shanghai Scientific Committee (grant no. 18060502600).

Acknowledgements. We thank Dr Zhenzhen for the assistance with experiment design.

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
