## [Reviewer comments · Royal Society Open Science]

Review History

RSOS-191478.R0 (Original submission)

Review form: Reviewer 1

Is the manuscript scientifically sound in its present form?

No

Are the interpretations and conclusions justified by the results?

No

Is the language acceptable?

Yes

Do you have any ethical concerns with this paper?

No

Have you any concerns about statistical analyses in this paper?

No

Recommendation?

Major revision is needed (please make suggestions in comments)

Comments to the Author(s)

The manuscript presents the combustion characteristics of flammable refrigerants leakage from heat pump system. The article needs a major revision.

1. The authors at many instances just presented the facts observed in the experiment rather than providing a detailed scientific reasoning. This needs to be improved. Also, the novelty of the current study and knowledge-gap compared to previous studies on same subject needs to be provided.
2. The authors have compared three refrigerants for ACHP system in electric vehicles. How results are useful in case of electric vehicles needs to be presented.
3. Page 9 Line 44: "It is interesting... 60 °C". Explanations can be presented rather than providing observations. Similar corrections need to be provided throughout the article.
4. Page 11. Line 44: "Considering the difficulty... experiment." What this mean?
5. Page 11. Line 50: "apparently" or "apparent"? Check the grammar, phrasing and English correctness throughout the manuscript.
6. Fig. 2 Give appropriate reference/source if you are using already published data or source.
7. Provide more clear images with high resolution for all figures.
8. Page 6 Line 13-15: Check correctness.
9. Table 1: Provide reference.

Decision letter (RSOS-191478.R0)

14-Jan-2020

Dear Dr Tu,

The editors assigned to your paper ("Experimental investigation on combustion characteristics of flammable refrigerant R290/R1234yf leakage from heat pump system for electric vehicles") have now received comments from reviewers. We would like you to revise your paper in accordance with the referee and Associate Editor suggestions which can be found below (not including confidential reports to the Editor). Please note this decision does not guarantee eventual acceptance.

Please submit a copy of your revised paper before 06-Feb-2020. Please note that the revision deadline will expire at 00.00am on this date. If we do not hear from you within this time then it will be assumed that the paper has been withdrawn. In exceptional circumstances, extensions may be possible if agreed with the Editorial Office in advance. We do not allow multiple rounds of revision so we urge you to make every effort to fully address all of the comments at this stage. If deemed necessary by the Editors, your manuscript will be sent back to one or more of the original reviewers for assessment. If the original reviewers are not available, we may invite new reviewers.

When submitting your revised manuscript, you must respond to the comments made by the

referees and upload a file "Response to Referees" in "Section 6 - File Upload". Please use this to document how you have responded to the comments, and the adjustments you have made. In order to expedite the processing of the revised manuscript, please be as specific as possible in your response.

- Data accessibility

If you wish to submit your supporting data or code to Dryad (<http://datadryad.org/>), or modify your current submission to dryad, please use the following link:
<http://datadryad.org/submit?journalID=RSOS&manu=RSOS-191478>

- Competing interests

- Authors' contributions

- Acknowledgements

- Funding statement

on behalf of Prof R. Kerry Rowe (Subject Editor)
openscience@royalsociety.org

Associate Editor's comments:

The Editors have made a recommendation based on the below and attached commentary. The view of the Editors is that the paper may be publishable after major revision. Please note the journal does not routinely permit multiple rounds of major revision. Please ensure you include the recommended changes from the reviewer and provide a full point-by-point response with your revision, too.

Comments to Author:

Reviewers' Comments to Author:
Reviewer: 1

Comments to the Author(s)

The manuscript presents the combustion characteristics of flammable refrigerants leakage form heat pump system. The article needs a major revision.

1. The authors at many instances just presented the facts observed in the experiment rather than providing a detailed scientific reasoning. This needs to be improved. Also, the novelty of the current study and knowledge-gap compared to previous studies on same subject needs to be provided.
2. The authors have compared three refrigerants for ACHP system in electric vehicles. How results are useful in case of electric vehicles needs to be presented.
3. Page 9 Line 44: "It is interesting... 60 °C". Explanations can be presented rather than providing observations. Similar corrections need to be provided throughout the article.
4. Page 11. Line 44: "Considering the difficulty... experiment." What this mean?
5. Page 11. Line 50: "apparently" or "apparent"? Check the grammar, phrasing and English correctness throughout the manuscript.
6. Fig. 2 Give appropriate reference/source if you are using already published data or source.
7. Provide more clear images with high resolution for all figures.
8. Page 6 Line 13-15: Check correctness.
9. Table 1: Provide reference.

Reviewers' Comments to Author:
Reviewer: 2

The novelty of this paper and the advance over the many other papers on the topic is not clearly elaborated. Without demonstrated novelty the paper should be rejected. Also while they provide data, its interpretation is rather limited and inadequate. The rationale behind their interpretation of the data needs to be more clearly defined.

Author's Response to Decision Letter for (RSOS-191478.R0)

See Appendix A.

RSOS-191478.R1 (Revision)

Review form: Reviewer 1

Is the manuscript scientifically sound in its present form?

Yes

Are the interpretations and conclusions justified by the results?

Yes

Is the language acceptable?

Yes

Do you have any ethical concerns with this paper?

No

Have you any concerns about statistical analyses in this paper?

No

Recommendation?

Accept as is

Comments to the Author(s)

Authors have reflected all the suggested comments/suggestions. I am recommending the manuscript in its revised present form.

Decision letter (RSOS-191478.R1)

23-Mar-2020

Dear Dr Tu,

It is a pleasure to accept your manuscript entitled "Experimental investigation on combustion characteristics of flammable refrigerant R290/R1234yf leakage from heat pump system for electric vehicles" in its current form for publication in Royal Society Open Science. The comments of the reviewers who reviewed your manuscript are included at the foot of this letter.

Please ensure that you send to the editorial office an editable version of your accepted manuscript, and individual files for each figure and table included in your manuscript as soon as possible. You can send these in a zip folder if more convenient. Failure to provide these files may delay the processing of your proof. You may disregard this request if you have already provided these files to the editorial office.

You can expect to receive a proof of your article in the near future. Please contact the editorial office (opscience_proofs@royalsociety.org) and the production office

(openscience@royalsociety.org) to let us know if you are likely to be away from e-mail contact -- if you are going to be away, please nominate a co-author (if available) to manage the proofing process, and ensure they are copied into your email to the journal.

Best regards,

on behalf of the Associate Editor and Professor R. Kerry Rowe (Subject Editor)
openscience@royalsociety.org

Associate Editor Comments to Author:

The reviewer who provided comments in the first round now considers the paper to be acceptable for publication.

Reviewer comments to Author:

Reviewer: 1
Comments to the Author(s)

Authors have reflected all the suggested comments/suggestions. I am recommending the manuscript in its revised present form.

Follow Royal Society Publishing on Twitter: [@RSocPublishing](https://twitter.com/RSocPublishing)

Appendix A

Dear Editor

Thank you for your letter and the reviewers' comments concerning our manuscript entitled "Experimental investigation on combustion characteristics of flammable refrigerant R290/R1234yf leakage from heat pump system for electric vehicles". We have studied these comments carefully and also have tried our best to make corrections, **please see the revised manuscript "Revision.docx"**, and revised portions are marked in red in an additional file (supplementary file) "**Revision-marked in red.docx**". The main corrections in the paper and the responses to the reviewer's comments are as following:

Response to Reviewers

Reviewer #1

Comment 1 - The authors at many instances just presented the facts observed in the experiment rather than providing a detailed scientific reasoning. This needs to be improved. Also, the novelty of the current study and knowledge-gap compared to previous studies on same subject needs to be provided.

Response:

Thanks for the Reviewer's comment. We have revised or enhanced the related portions of manuscript for a better understanding (please see detail in Introduction and Section 3 as marked in red).

As described in Introduction, previous works mainly focused on the risk assessment and combustion characteristics of R290/R1234yf in household air conditioning system. Literatures about R290 or R1234yf combustion behavior during leakage process from ACHP system in EVs or ICEVs are still scarce. Considering the environment of application, the ACHP of vehicles would show significant differences with (much more complicated than) household AC. That is why we wish to conduct this work based on our newly designed test facility.

The novelty of our work includes theoretical and experimental study on various initial gas temperature and volume flow rate effects on combustion characteristics of flammable refrigerant R290/R1234yf leakage from heat pump system for electric vehicles, which was validated by the experimental data. We also added these descriptions in our revised manuscript to enhance our novelty in Introduction Section as:

"For further application of R290 and R1234yf refrigerant in EVs, leakage and combustion risk assessment need to be clarified. R290 and R1234yf were classed as A3 and A2L flammable refrigerant respectively by American society of heating, refrigerating and air-conditioning engineers (ASHARE) [24]. Under normal temperature conditions, R290 could be easily ignited **causing** potential fire or explosion hazard of R290 from ACHP system **when** the unpredictable leaked R290 reach lower flammable limit (LFL). Previous study on leakage behaviors of R290 from household air conditioning showed that the fire hazard could probably occur in the early stage of the leakage process near leak source [25]. Feng et al. have **studied** the combustion and explosion characteristics of R290 and R1234yf near LFL [26, 27], **which focused on the effect of**

gas disturbance and combustion intensity, but without considering the effect of other initial condition such as gas temperature. Clodic and Jabbour have reported the burning rates of R290 using a tube experimental method for household air conditioner [28]. We noted that there are relative studies on the combustion characteristics of R290 leakage for household ACs, but few researches were involved for electric vehicle. Further, a series of combustion experiments were carried out by Zhang et al. in laboratory scale, which indicates that the over-pressure leaked R290 is not sufficient to cause AC system damage, but if R290 was ignited during the leak, the system would quickly be burned out [29].

Comparing with the risk assessment and combustion investigation of flammable refrigerants for household AC system, literatures about R290 or R1234yf combustion behavior during leakage process from ACHP system in EVs or ICEVs are still limited. In this paper, a newly designed ACHP system, applied in a type of EVs, was introduced. Based on this system, thermodynamic characteristics were studied using R290 and R1234yf as alternative refrigerant of R134a. Then, a comprehensive experimental study using R134a/R290/R1234yf with various initial temperature and volume flow rate were conducted under operation conditions. Typical combustion and thermal dynamics parameters including flame height and centerline temperature of flame under various conditions were analyzed. Ignition temperature and radiation flux were also recorded to investigate the combustion behavior of the R290/R1234yf flame. Finally, the combustion behavior of these flammable refrigerants were compared and followed with some conclusive remarks. The data obtained could provide reference for the electric vehicles safety design with R290/R1234yf ACHP system.”

Comment 2 - The authors have compared three refrigerants for ACHP system in electric vehicles. How results are useful in case of electric vehicles needs to be presented.

Response:

As the Reviewer said, in this paper, 3 typical refrigerants were selected for comparison tests, and also combustion characteristics are measured under different leakage mass flow rate and gas temperature conditions.

In the ACHP system in EVs/ICEVs, fire hazard is one potential threat for the safety of vehicles. So we chose some key parameters including ignition temperature or energy, flow temperature field, flame shape and radiation, etc. to investigate the combustion behaviors of these flammable refrigerants under complex conditions. These results were believed to be useful for the electric vehicles safety design and fire protection or rescue with R290/R1234yf ACHP system (and further work will be conducted on leakage combustion behaviors of other refrigerants, e.g., R1234zd and R1234ze.).

According to the comment, we added some new parts in Conclusion Section to clarify the usage as:

“(1) Comparing with mass flow rate, the gas temperature showed little effect on flame height

of R290, while the combustion intensity is enhanced with high initial temperature. The flame from R290 burning is wider than R1234yf for the same initial condition in general. The flame Physical morphology obtained here could provide guidance for fire hazard evaluation and rescue of vehicles.

(2) It was found that the R290 under initial temperature 10 °C burned vigorously, and for 90 °C, 1.5 L/min condition, R290 almost blowing out. On the other hand, R1234yf could not be ignited without flaming fire, and also accompanied with black smoke and pungent smell. Moreover, flame height showed a positive relation with refrigerant volume flow rate in general.

(3) Ignition temperatures are usually above 600 °C for R290 and keep approximately steady except for high initial gas temperature condition. Further, radiation flux increased significantly with enlarged flow rate for R290, which is different from the results of R1234yf. These tendencies obtained were believed to be useful for early fire detection and prevention system design inside new energy cars.

Further work will be conducted on leakage combustion behaviors of other refrigerants, e.g., R1234zd and R1234ze.”

Comment 3 - Page 9 Line 44: “It is interesting... 60 °C”. Explanations can be presented rather than providing observations. Similar corrections need to be provided through out the article.

Response:

We have checked the original manuscript and found that there were some mistakes in the expression of this sentence. According to the Reviewers good suggestion, we also added some for a better expression. The sentence has been revised and added as follow, please also see the revised manuscript in Section 3.1.

“Under volume flow rate 1.25L/min, R1234yf got the highest flame height when gas heated to initial temperature 90 °C by observation. It is the competitive result by effects of burning enhancement by temperature, flow turbulence or destabilization by exit velocity. Among three temperature conditions, flame height of R290 is approximately unchanged, while combustion intensity seems to be enhanced with the increasing gas temperature from flame color. As the temperature rises, the color of the flame ranges from red and orange to yellow and white.”

Also, according to the good suggestion, we have enhanced the scientific explanation through out the article especially in Section 3.

Comment 4 - Page 11. Line 44: “Considering the difficulty... experiment.” What this mean?

Response:

Sorry for the confusing sentence here, it means that the ignition conditions for refrigerant R1234yf cannot be easily achieved. SAE international conducted extensive tests on the toxicity and flammability of R1234yf, the result showed that the R1234yf had low flammability, but

needed to be ignited in the presence of gasoline (flammability similar to R134a).

Since the sentence is not clearly expressed, we have revised it as “Consider the difficulty of igniting the refrigerant R1234yf” in Section 3.2.

Comment 5 - Page 11. Line 50: “apparently” or “apparent”? Check the grammar, phrasing and English correctness throughout the manuscript.

Response:

We have replaced the word “apparently” with “apparent” as the reviewer’s suggestion. Also we have checked the whole manuscript to avoid grammar errors.

Comment 6 - Fig. 2 Give appropriate reference/source if you are using already published data or source.

Response:

Fig.2 was calculated by “NIST Refprop” using the experimental conditions in Tables 2 and 3. We have added the reference by the good suggestion.

Comment 7 - Provide more clear images with high resolution for all figures.

Response:

Thanks for the Reviewer’s comments. All the figures in the original manuscript have been substituted by high-resolution figures (An example is showed as follow).

Fig. 9. Flame temperature of R290 alone centerline in burning under simulated leakage.

Comment 8 - Page 6 Line 13-15: Check correctness.

Response:

Thanks for the valuable comments from reviewer. We have checked it and corrected as “To figure out the risk of the R290 utilizing on an automobile ACHP system, three heat exchanger-ACHP system with functions of cooling and heating was designed as shown in Fig. 1”. Please see detail in the revised manuscript Section 2.2.

Comment 9 - Table 1: Provide reference.

Response:

We have added the reference for Table 1 as: [30] ANSI/ASHRAE Standard 34-2010 Designation and Safety Classification of Refrigerants. Atlanta GA, 2010.

Reviewer #2

Comment 1 - The novelty of this paper and the advance over the many other papers on the topic is not clearly elaborated. Without demonstrated novelty the paper should be rejected. Also while they provide data, its interpretation is rather limited and inadequate. The rationale behind their interpretation of the data needs to be more clearly defined.

Response:

We appreciate the Reviewer’s comment and apologize for inexplicit illustration of innovation in manuscript, and we tried our best to revise for a better understanding. Similar to the response for Reviewer 1, the novelty of our work includes theoretical and experimental study on various initial gas temperature and volume flow rate effects on combustion characteristics of flammable refrigerant R290/R1234yf leakage from heat pump system for electric vehicles, which was validated by the experimental data.

Previous works mainly focused on the risk assessment and combustion characteristics of R290/R1234yf in household air conditioning system. Literatures about R290 or R1234yf combustion behavior during leakage process from ACHP system in EVs or ICEVs are still scarce. Considering the environment of application, the ACHP of vehicles would show significant differences with (much more complicated than) household AC. That is why we wish to conduct this work based on our newly designed test facility. We also added these descriptions in our revised manuscript to enhance our novelty in Introduction Section as:

“For further application of R290 and R1234yf refrigerant in EVs, leakage and combustion risk assessment need to be clarified. R290 and R1234yf were classed as A3 and A2L flammable refrigerant respectively by American society of heating, refrigerating and air-conditioning engineers (ASHARE) [24]. Under normal temperature conditions, R290 could be easily ignited causing potential fire or explosion hazard of R290 from ACHP system when the unpredictable

leaked R290 reach lower flammable limit (LFL). Previous study on leakage behaviors of R290 from household air conditioning showed that the fire hazard could probably occur in the early stage of the leakage process near leak source [25]. Feng et al. have studied the combustion and explosion characteristics of R290 and R1234yf near LFL [26, 27], which focused on the effect of gas disturbance and combustion intensity, but without considering the effect of other initial condition such as gas temperature. Clodic and Jabbour have reported the burning rates of R290 using a tube experimental method for household air conditioner [28]. We noted that there are relative studies on the combustion characteristics of R290 leakage for household ACs, but few researches were involved for electric vehicle. Further, a series of combustion experiments were carried out by Zhang et al. in laboratory scale, which indicates that the over-pressure leaked R290 is not sufficient to cause AC system damage, but if R290 was ignited during the leak, the system would quickly be burned out [29].

Comparing with the risk assessment and combustion investigation of flammable refrigerants for household AC system, literatures about R290 or R1234yf combustion behavior during leakage process from ACHP system in EVs or ICEVs are still limited. In this paper, a newly designed ACHP system, applied in a type of EVs, was introduced. Based on this system, thermodynamic characteristics were studied using R290 and R1234yf as alternative refrigerant of R134a. Then, a comprehensive experimental study using R134a/R290/R1234yf with various initial temperature and volume flow rate were conducted under operation conditions. Typical combustion and thermal dynamics parameters including flame height and centerline temperature of flame under various conditions were analyzed. Ignition temperature and radiation flux were also recorded to investigate the combustion behavior of the R290/R1234yf flame. Finally, the combustion behavior of these flammable refrigerants were compared and followed with some conclusive remarks. The data obtained could provide reference for the electric vehicles safety design with R290/R1234yf ACHP system.”

Moreover, according to the good comment, we have enhanced the scientific explanation through out the article especially in Section 3 (Please see detail in the revised manuscript).